# Beyond Impairment of Virion Infectivity: New Activities of the Anti-HIV Host Cell Factor SERINC5

**DOI:** 10.3390/v16020284

**Published:** 2024-02-12

**Authors:** Samy Sid Ahmed, Kathrin Bajak, Oliver T. Fackler

**Affiliations:** 1Department of Infectious Diseases, Integrative Virology, University Hospital Heidelberg, Im Neuenheimer Feld 344, 69120 Heidelberg, Germany; samy.sidahmed@med.uni-heidelberg.de (S.S.A.); kathrin.bajak@med.uni-heidelberg.de (K.B.); 2German Centre for Infection Research (DZIF), Partner Site Heidelberg, 38124 Heidelberg, Germany

**Keywords:** SERINC5, HIV-1, restriction factor, innate immunity, lipid asymmetry

## Abstract

Members of the serine incorporator (SERINC) protein family exert broad antiviral activity, and many viruses encode SERINC antagonists to circumvent these restrictions. Significant new insight was recently gained into the mechanisms that mediate restriction and antagonism. In this review, we summarize our current understanding of the mode of action and relevance of SERINC proteins in HIV-1 infection. Particular focus will be placed on recent findings that provided important new mechanistic insights into the restriction of HIV-1 virion infectivity, including the discovery of SERINC’s lipid scramblase activity and its antagonism by the HIV-1 pathogenesis factor Nef. We also discuss the identification and implications of several additional antiviral activities by which SERINC proteins enhance pro-inflammatory signaling and reduce viral gene expression in myeloid cells. SERINC proteins emerge as versatile and multifunctional regulators of cell-intrinsic immunity against HIV-1 infection.

## 1. Introduction: SERINC Proteins as Antiviral Restriction Factors

Our immune system comprises a number of defense mechanisms to protect us from pathogenesis induced by infections with viruses such as human immunodeficiency virus (HIV). These include adaptive responses, e.g., the lysis of infected cells by cytotoxic T or natural killer cells, as well as humoral responses aimed at generating neutralizing antibodies [1,2]. While these mechanisms, in principle, very potently limit virus replication and spread, HIV has developed efficient evasion mechanisms to bypass and suppress adaptive immune reactions at later stages of infection [3,4]. In the initial phase of infection and, thus, prior to the development of the adaptive response, innate immune mechanisms provide a first line of defense that is of critical importance for fending off virus infection and for priming subsequent adaptive responses [5,6]. Key effectors of innate immunity to HIV infection include the cell-intrinsic ability to elicit signaling triggered by the sensing of viral structures (pathogen-associated molecular patterns, PAMPs) by pattern recognition receptors (PRRs). Such signaling induces antiviral states in infected and bystander cells and drives the expression of physical barriers to virus replication (so-called restriction factors, RFs). Restriction factors can be defined as a set of constitutively expressed proteins that directly impair various steps of viral replication cycles. The expression of many of these intrinsic immune actors is increased by interferons, and some can induce innate immune signaling upon recognition of viral components. Intense research in the past decades has identified that human cells dispose of an impressive array of RFs that collectively target virtually any step of the HIV life cycle, from virus entry and intracellular replication to the release of new viral progeny. Because of the collective antiviral potency of host cell RFs, the productive infection of such target cells by HIV requires specific countermeasures. RF antagonism is typically achieved through the expression of a viral antagonist that inactivates the RF, the ability to adapt the structure targeted by the RF to evade recognition at the low cost of replication efficacy or modulating the transcriptional activity of RF coding genes [7].

With Vpu, Vpr, Vif and Nef, HIV-1 encodes for four accessory proteins that are dispensable for virus growth in cell culture models that lack the expression of the RF they antagonize but which are important pathogenic determinants in vivo [8]. While the role of Vpu, Vpr and Vif as antagonists of prominent RFs such as tetherin, the HUSH complex and APOBEC3G is well established [5,9,10,11,12], no such antagonistic activity was identified for the Nef protein until 2015. Nef is a multifunctional protein interaction adaptor that increases viral load and disease progression in HIV-infected individuals, and in light of the plethora of host ligands identified [13,14], it was surprising that the viral pathogenesis factor remained an orphan antagonist for many years. In 2015, two parallel studies identified serine incorporator (SERINC) proteins 3 and 5 as host cell RFs that are antagonized by Nef [15,16,17,18]. Both studies compared related cell systems that displayed marked differences in the magnitude of HIV-1 infectivity enhancement caused by Nef. The analysis of cellular expression profiles and virion incorporation patterns identified SERINC5 (S5) and SERINC3 (S3) as antiviral proteins that dampen the infectivity of cell-free HIV particles and can be efficiently antagonized by HIV-1 Nef. Importantly, the ability to antagonize SERINC proteins is well conserved among Nef variants throughout lentiviral evolution [19]. Subsequent studies revealed that the suppression of virion infectivity is not limited to HIV-1 but that SERINC proteins also target other viruses, including murine leukemia virus (MLV) [20], equine infectious anemia virus (EIAV) [21], feline leukemia virus-B (FeLV-B) [22], influenza virus [23,24], classical swine fever virus [25] and SARS-CoV-2 [26]. The demonstration that S3 and S5 reduce the glycosylation of hepatitis B virus (HBV) glycoproteins to inhibit the secretion of virus particles [27] revealed that SERINC proteins can interfere with virus replication using multiple strategies. Most viruses that are sensitive to SERINC restriction express viral proteins for its antagonism (e.g., Nef for HIV-1; GlycoGag for MLV and FeLV-B; Orf7a for SARS-CoV-2; S2 for EIAV; small, large and mid-size surface proteins for HBV) [28]. Together, these findings suggest SERINC’s antiviral function as an important player in the tug-of-war between viruses and their hosts and have triggered intense research efforts aimed at dissecting the mechanisms of restriction and antagonism, as well as their physiological relevance.

## 2. SERINC Protein Organization and Evolution

Mammalian SERINC proteins comprise a family with five members that share a complex architecture. Originally predicted to contain eleven transmembrane domains [29], solving the cryo-EM structure of the Drosophila melanogaster SERINC protein TMS1 revealed ten transmembrane alpha-helix domains that are organized into two subdomains, which are connected by a long diagonal helix (helix 4). The solved structurealso displayed five extracellular loops (ECLs), four intracellular loops (ICLs) and a lipid-binding groove, and human S5 adopts an analogous structure [30] (Figure 1; also see Section 6).

*SERINC* genes are present in all eukaryotes, with the one gene found in, e.g., D. melanogaster and S. cerevisiae expanded to five in higher eukaryotes (see reviews by Firrito et al. and Cano-Ortiz et al. for more details on SERINC evolution [18,31]). The five mammalian SERINC proteins share their main domain organization and 31–58% amino acid identity. Human S3 and S5 were identified as the most potent inhibitors of HIV-1 infectivity, and SERINC 1 (S1) and 4 (S4) can also impair HIV-1 infectivity. In contrast, SERINC 2 (S2) lacks anti-HIV activity [32,33]. Importantly, rodent, lagomorph and feline orthologs of S5 and S3 restrict HIV virion infectivity [31,34]. Consistent with this evolutionary conservation, S3 and S5 did not evolve in response to the introduction of the pathogens they antagonize in the human population [35]. This lack of an evolutionary footprint suggests that the antiviral activities of SERINC proteins are an intrinsic property acquired early in evolution, and in fact, even the coelacanth S2 protein bears anti-HIV activity [33].

## 3. Expression and Role of SERINC Proteins in HIV Pathogenesis

The broad antiviral activity of SERINC proteins; the frequent development of viral antagonists; and the evolutionary conservation of these features suggest that they play an important role in virus pathogenesis. Most studies of the antiviral mechanism rely on the characterization of overexpressed SERINC variants in cell systems that lack the significant expression of endogenous SERINC mRNA, but an important insight into SERINC protein function comes from gene silencing or knock-out approaches to suppress the expression of endogenous proteins. Importantly, in the CD4 T cell lines that defined S5 as a key factor for virion infectivity restriction antagonized by Nef, the suppression of S5 expression confers a replication advantage to *nef*-deficient HIV-1 [15,16]. Maybe the most striking evidence for the functional relevance of the S5 restriction of virus spread in vivo comes from the MLV mouse system, where the knock-out of the S5 gene overcomes the pronounced replication deficit of MLV variants that lack the S5 antagonist GlycoGag [36]. In line with such a role for S5 in controlling viral spread in vivo, the ability of SIV Nef variants engineered to lack S5 antagonism to boost SIV replication in vivo is significantly reduced [37]. Several observations from patient cohorts and different natural hosts of lentivirus infection also suggest a crucial role for S5: the magnitude by which lentiviral Nefs can antagonize S5 correlates with the prevalence of the corresponding lentiviral lineage [19]; correlations were reported for the extent of S5 antagonism with clinical progression and viral load [38,39,40,41], and levels of SERINC mRNA have been found to be decreased in HIV patients [42]. On the other hand, Nef can boost HIV-1 replication in the absence of SERINC proteins in some cell systems [43,44], and an SIV Nef variant that potently targets human S5 for degradation does not facilitate HIV-1 spread in primary human CD4 T cells [45]. While these partially contradictory findings imply that the antiviral activities of SERINC proteins depend on the cellular context, a confounding problem in this field remains that the detection of SERINC proteins expressed at endogenous levels is difficult, and physiological protein expression distribution and levels remain unknown. Importantly, the expression of an epitope-tagged S5 protein from the *S5* locus provides antiviral activity [46]. While this finding demonstrates that S5 protein levels below the limit of detection with the currently available tools can be functional, it remains unclear in which cells the S5 protein is expressed physiologically. In principle, the expression of SERINC mRNA can be detected in all human and mouse tissues [18,22]. A comprehensive characterization of SERINC mRNA levels specifically in primary human HIV target cells [47] revealed that expression is moderate and not responsive to cell activation or interferons in CD4 T cells, which may explain why the depletion of S5 does not affect HIV-1 spread in CD4 T cells (see above). In myeloid cells, in turn, S5 expression is low in undifferentiated monocytes but significantly increases upon differentiation into, e.g., macrophages [47]. More sensitive tools for the detection of endogenously expressed SERINC proteins would be an asset in the dissection of the pathophysiological role of S5.

## 4. Mechanism of S5 Restriction of HIV-1 Virion Infectivity

The initial studies that identified the antiviral activity of S5 and S3 established virion fusion with target cells as the main step in the viral life cycle that is impaired by the restriction (Figure 2). This observation triggered a series of subsequent studies that focused on the impact of SERINC proteins and the function of the HIV-1 glycoprotein Env. These analyses established that S5 impairs several steps of the membrane fusion process, including hemifusion, fusion pore opening and fusion pore dilation [48,49]. Mechanistic studies have revealed that these functional defects are associated with a reduction in the organization of Env into clusters in virions, a requirement for full infectivity [50], and different resistance to cholesterol extraction [51]. Collectively, these effects may determine an impairment of Env function that is characterized by an altered conformation and sensitivity to antibody-mediated neutralization [49,52,53,54,55,56] (Figure 2). Interestingly, not all HIV-1 Env variants are equally sensitive to the virion infectivity restriction caused by SERINC proteins [15,16]. HIV-1 Env is composed of gp120, which provides surfaces for interactions with entry receptors that are organized into five variable loops (V1–V5) and gp41, which provides fusion activity. Determinants for resistance to restriction caused by S5 have been identified in the V1, V2 and V3 loops of gp120 and in the cytoplasmic tail of gp41 [16,57,58], and the sensitivity to the S5 restriction differs between patient-derived Env proteins [59]. Interestingly, the nature of the viral core incorporated into lentiviral particles also affects the sensitivity to restriction caused by S5 [60], and defining the molecular basis of these dependencies will provide important new insights into the structure–function relationship of S5 restriction.

Parallel efforts have been aimed at determining functional S5 domains and motifs. A structure–function analysis by Pye et al. [30] identified ECL3 and ECL5 and, in particular, a 412FNYESANIE420 motif in ECL5 as important antiviral features of the RF. The relevance of an aromatic residue at position 412 for restriction was further underscored by a study by Tan et al. [61] (Figure 1). In addition, residues in the transmembrane domains that form the lipid-binding groove of S5 are also required for its anti-HIV activity [30] (Figure 1; see also Section 6).

## 5. Mechanism of Nef-Mediated Antagonism of S5 Restriction of HIV-1 Virion Infectivity

The two studies that identified the anti-HIV activity of SERINC proteins described the exclusion of RFs from virions as an active principle of Nef antagonism, which subsequently was found to also be employed by other viral antagonists such as EIAV S2 and MLV GlycoGag [15,16,62,63]. One of Nef’s main activities is to reduce the cell surface exposure of a broad range of host cell surface receptors, and the cell surface levels of SERINC proteins can indeed be reduced by the expression of Nef [64,65,66]. Conceivably, lowering the cell surface levels of SERINC proteins results in reduced virion incorporation, and this mechanism is considered the main mechanism of Nef antagonism to the virion infectivity restriction caused by S3 and S5 [22,67]. It is, however, noteworthy that (i) Nef-mediated reduction in S5 cell surface exposure and the concomitant lysosomal degradation of the RF has not been observed in all cell systems in which Nef antagonizes S5 restriction [15,68]; (ii) some Nef mutants that antagonize S5 fail to reduce its cell surface exposure and the magnitude of S5 antagonism and internalization caused by patient-derived variants do not always correlate [38,68]; and (iii) the Nef protein of SIVcol evolved a mechanism to target S5 for proteasomal degradation but does not enhance HIV replication [45]. Moreover, Nef also increases the infectivity of virions that incorporate S5 [16,68]. Since analyses of S5 incorporated into virions have been conducted in bulk, this may reflect that Nef increases the fraction of particles that carry low levels of S5 that are not sufficient to impair their infectivity. Alternatively, these findings suggest that Nef, a protein that is efficiently packaged into HIV-1 virions itself [69,70], may also antagonize the activity of virion-associated S5 (Figure 2).

Significant progress was made toward understanding the mechanism of Nef antagonism by reducing cell surface exposure of S5. The initial mapping of determinants in Nef required for antagonism revealed the critical role of membrane association, mediated by N-terminal myristoylation at glycine 2 of Nef, as well as an overlap with endocytic motifs required for the internalization of cell surface CD4, in particular, a di-leucine interaction motif for endocytic adaptor complex 2 (AP2) [15,16,68] (Figure 3). Consistent with the role of the Nef-mediated internalization of S5 via the AP-2 pathway caused by dimeric Nef, expression of the endocytosis-driving GTPase dynamin 2 and presence of residues required for Nef’s interactions with dynamin 2 (D123, an acidic residue at position 150 of Nef that is involved in interactions with AP-2 and the dimer interface of Nef (L112, Tyr 115, F121)) are essential for S5 antagonism by the HIV-1 pathogenesis factor [15,71,72]. Patient-derived Nef variants, however, have revealed that subtle differences exist between the contribution of individual residues within the di-leucine motif for CD4 downregulation and S5 antagonism [73,74]. Consistently, subsequent studies have revealed that the molecular machinery that Nef hijacks to reduce the cell surface exposure of S5 differs from the classical clathrin-AP-2 internalization pathway employed for CD4. S5 antagonism requires the ICL4 in the RF to act as a central element of its sensitivity to antagonism by Nef anti-HIV activity (Figure 1) [75]. ICL4 not only contains an LxL motif at position 350–352 [75,76] that is critical to Nef antagonism but also a cluster of acidic residues that reduces the ability of Nef to counteract S5 [76]. In addition, Chai et al. showed that Nef physically associates with S5 to recruit Cyclin K and cyclin-dependent kinase 13 [77]. This results in the phosphorylation of serine 360 in S5’s ICL4 and the subsequent AP-2-dependent internalization of S5 in Rab7+ endosomes and lysosomal degradation. A recent study by Li and colleagues [78] identified the ubiquitination of lysine 130 in S5 caused by the E3 ubiquitin ligase Cul3-KLHL20 as an important regulator of both its targeting to the plasma membrane and its internalization and degradation. While K33-linked ubiquitination is involved in the plasma membrane targeting of S5, the K48-linked ubiquitination of S5 facilitates its internalization and, thus, Nef-mediated antagonism. These studies define K33/K48-linked polyubiquitination as a novel mechanism used to regulate cell surface receptor trafficking in model cell lines. It will be interesting to assess if the use of this mechanism is a shared feature of cell surface receptors whose internalization is triggered by Nef and how it is applied in physiological HIV target cells in which S5 restricts HIV-1 virion infectivity. Moreover, a stretch of amino acids in the flexible N-terminal anchor domain of Nef (the S5-antagonism motif, S5AM) has been identified as an additional amino acid motif in Nef that contributes to S5 antagonism but still awaits the identification of the responsible host cell ligands. The S5AM is located in proximity to the binding site of the Nef-associated kinase complex [79] but mediates distinct yet unknown protein interactions [80], and the analogous motif in simian immunodeficiency virus (SIV) Nef is important for promoting virus replication in primary CD4 T cells [37]. In addition, an analysis of patient-derived HIV-1 Nef variants indicated roles for residues K94 and H116 in S5 antagonism [40].

## 6. Phospholipid Scramblase Activity of SERINC Proteins

SERINC proteins were termed serine incorporators based on the initial publication by Inuzuka and colleagues, who reported that members of the protein family incorporate the amino acid serine into lipid biosynthesis to promote the production of serine-derived lipids such as phosphatidylserine (PS) and sphingolipids [29]. Since PS is enriched in HIV particles and represents an important determinant of virion infectivity [81], we determined whether the expression of S5 during virus production alters the lipid composition of HIV-1 particles [82]. Quantitative lipidomics did not reveal an impact of S5 on the lipid composition, including levels of PS, HIV-producing cells or virus particles. A flow cytometry-based analysis of PS levels on the virion surface also did not provide evidence of S5-mediated alterations. Similarly, the knock-out of *SERINC1* in mice did not alter serine-derived lipid composition [83], and collectively, these results suggested that SERINC proteins do not regulate lipid biosynthesis. The cyro-EM structure of TMS1, however, has revealed a lipid-binding groove [30] (Figure 1), and a recent study by Leonhard and colleagues showed that SERINC proteins can alter the lipid asymmetry of membranes [84]. Solving the cryoEM structure of human S3 revealed the strong structural similarity of SERINC proteins to unregulated lipid transporters. Consistently, the reconstitution of S3 into proteoliposomes demonstrated its ability to translocate lipids from the inner to outer membrane leaflet (scramblase activity). This effect was not limited to PS but also included the flipping of phosphatidylcholine (PC) and phosphatidylethanolamine (PE) [84] and may, therefore, explain the observation that the S5-mediated restriction of membrane fusion involves local alterations of lipid order and heterogeneity [85]. Single-virion microscopy, as well as flow cytometry of complexed virus particles, demonstrated that the presence of S5 during particle production markedly increases PS levels on the virion surface. Extending previous studies reporting the effects of S5 on the conformation of HIV-1 Env [54,55], single-molecule FRET analyses have revealed that the virion incorporation of S5 drives the viral glycoprotein Env into a conformational state, with lower potential to mediate fusion with target membranes. Similar reductions in infectivity and alterations in Env confirmation were observed upon virion incorporation of the unrelated phospholipid scramblase TMEM16F. Finally, Nef partially antagonized an increase ofn PS exposure at the surface of HIV virions and the ability of S5 mutants to promote PS flipping, and to restrict HIV infectivity correlated well. While the mechanism by which Nef counteracts PS externalization in HIV-1 particles produced in the presence of S5 is unknown, virion exclusion of the restriction factor is an intuitive explanation. In addition, the inhibition of the lipid scramblase activity of S5 in virus particles caused by virion pools of Nef may account for the inhibition of S5 molecules in HIV particles [69,86]. Together, these results provide strong evidence that the newly discovered phospholipid scramblase activity of SERINC proteins rather than the previously assumed alterations in lipid biosynthesis is critical for the restriction of HIV particle infectivity.

## 7. S5 Activities in HIV-1 Infection beyond Reducing Virion Infectivity

In addition to the impairment of virion infectivity, recent studies have identified three additional antiviral activities of S5 in monocyte-derived macrophages. Zeng et al. reported that the expression of S3 and S5 increases antiviral signaling [87] (Figure 4, right). Reducing the expression of each of these RFs decreased the NF-kB-mediated expression of pro-inflammatory cytokines in response to potent innate immune triggers such as LPS, polyI:C and the Sendai virus. Consistently, overexpression of S3 or S5 increased NF-kB signaling in HEK293T cells. Mechanistic dissection revealed that SERINC proteins associate with the innate signaling adaptor proteins MAVS and TRAF6, which coordinate signaling downstream of the dsRNA PRRs RIG-I and MDA5. This is in line with a recent report that showed that S5 interacts with MDA5 to promote type-I interferon signaling [25]. This association may be instrumental for the ability of S5 to promote pro-inflammatory signaling, as functional experiments have indicated that S5 expressed in target cells contributes to innate immune signaling induced by virus infection and to the induction of an antiviral state. These results suggest that, similarly to the RFs Trim5α and tetherin [88,89], S5 promotes antiviral signaling. While signaling is triggered by Trim5α and tetherin upon the recognition of viral capsids and budding structures, respectively, S5 appears to act as an integral part of the MAVS/TRAF6 pathway. Since the physiological trigger of this pathway (dsRNA) is not a natural product of the HIV life cycle, the role of S5 may be more relevant in the context of infections with dsRNA viruses.

While these effects on pro-inflammatory signaling are mediated by S5 expressed in the target cells of virus infections, we observed that the presence of S5 during HIV-1 particle production also sensitizes these virions for innate immune recognition by primary monocyte-derived macrophage (MDM) target cells. Similar to the effects of S5 expression in target cells, this effect resulted in the increased secretion of pro-inflammatory cytokines such as IL-6 and TNFα [90] (Figure 2, right). This sensitization for innate immune recognition occurred in the absence of detectable effects of S5 on rates of productive infection; was antagonized by the presence of Nef during particle production; and did not require the productive infection of the MDMs. While the PAMPs and PRRs involved remain to be determined, these results suggest that S5 can increase the recognition of HIV-1 particles by innate immune sensors during the non-productive uptake of virus particles. This may reflect that virion-associated S5 targets virus particles to an uptake pathway that allows for particularly efficient sensing or alters particle architecture in a manner that facilitates innate immune recognition. In an alternative and intriguing scenario, virion-associated S5 may, upon the uptake of these particles, mediate the local activation of MAVS/TRAF6, as described by Zeng et al., for S5 expressed in MDMs to induce antiviral, pro-inflammatory signaling [87].

**Figure 4 viruses-16-00284-f004:**
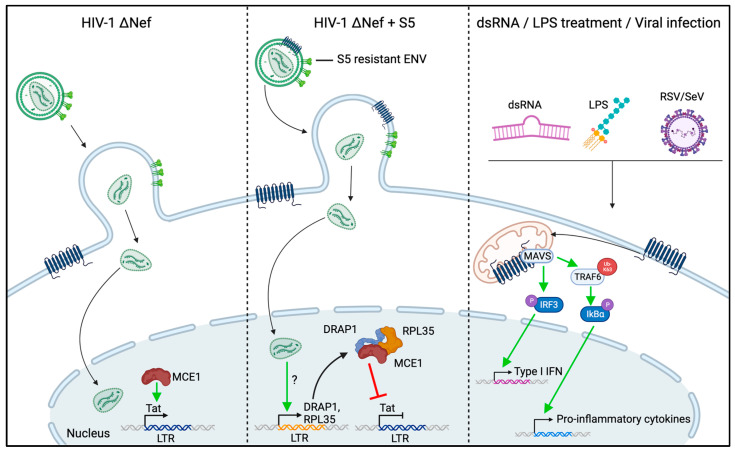
Schematic summarizing recent findings on the effects of SERINC5 expression on myeloid HIV-1 target cells. The left and middle panels depict how, under conditions where S5 does not impair HIV-1 infectivity, virion-associated pools of the RF can reduce the transcription of HIV-1 genes by interfering with the activity of the viral transactivator Tat [91]. The right panel depicts the effects of endogenous S5 on innate signaling via MAVS [87]. See text for more details. The figure was generated using Biorender.

More recently, yet another consequence of the challenge of myeloid HIV target cells with S5-containing HIV-1 particles was described. Ramdas and Chande showed that even when HIV-1 Env variants that are not sensitive to infectivity restriction caused by S5 are used for infection, the incorporation of S5 during particle production impacts the rates of productive infection in myeloid but not lymphoid cells [92] (Figure 4, left and middle). This effect maps to the viral gene expression of integrated proviral copies, as well as to viral mRNA export, and is associated with the induced expression of the mRNA-binding protein RPL35 and the transcriptional repressor DRAP1. Knock-out studies have revealed that the depletion of both of these proteins alleviates reductions in productive infection caused by S5. Interestingly, the authors identified that both RPL35 and DRAP1 are recruited into a complex with the mammalian-capping enzyme MCE1 in cells challenged with S5-containing particles, and the depletion or overexpression of MCE1 phenocopied the corresponding effects of RPL35 and DRAP1. MCE1 is known to interact with and support the function of the viral transcriptional activator Tat [93], and Ramdas and Chande demonstrated that the increased levels of RPL35/DRAP efficiently outcompete Tat for interactions with MCE1, thereby reducing Tat activity in viral gene expression. In line with this study and the proposed Tat-dependent mechanism, Shi et al. observed in non-myeloid HEK293T cells and Jurkat cells that S5 reduces viral but not cellular or ectopic gene expression and limits the efficacy ofproviral DNA integration [91]. It will be interesting to see if these antiviral effects and their antagonism by Nef are also exerted in primary HIV target cells. Taken together, these recent studies reveal that S5 has important functions in addition to impairing virion infectivity. Most of these novel activities appear to be specific to myeloid cells and can be elicited either by virion-associated S5 or its expression in target cells. Their physiological relevance and contribution to the effects of S5 on HIV-1 spread, as well as the involvement of S5′s lipid scramblase activity, warrant further investigation.

## 8. Conclusions and Open Questions

S5 emerges as a potent host cell RF that dampens the infectivity of cell-free virus particles. Significant progress was made in our understanding of the mechanisms by which S5 reduces HIV-1 infectivity and HIV-1 Nef antagonizes this restriction. The recent discovery that S3 and S5 exert lipid-scrambling activity to elevate the levels of PS on the virion surface significantly adds to this understanding. In parallel, new effector functions of S5 in triggering antiviral signaling; sensitizing viral particles for innate immune recognition; and regulating HIV gene expression have emerged. A key goal will now be to define to what extent these effector functions are conserved among SERINC family members. If S5’s scramblase activity contributes to these effector functions, addressing which mechanism alterations in lipid asymmetry impact these activities will be another important goal. Moreover, S5 activities have, so far, been exclusively studied in the context of infections with cell-free viruses. Since cell-associated modes of virus transmission play an important role in HIV-1 spread and provide a means to circumvent some host cell restrictions [94,95,96,97], it will be important to address the impact of S5 in the context of cell–cell transmission. Finally, a main challenge lies in the visualization of the S5 protein and the characterization of its function in a physiological context. The precise definition of the physiological target cell and the activation/differentiation state by which S5 exerts its antiviral roles may also open avenues for the identification of the physiological roles of SERINC proteins beyond the restriction of virus spread.

## Figures and Tables

**Figure 1 viruses-16-00284-f001:**
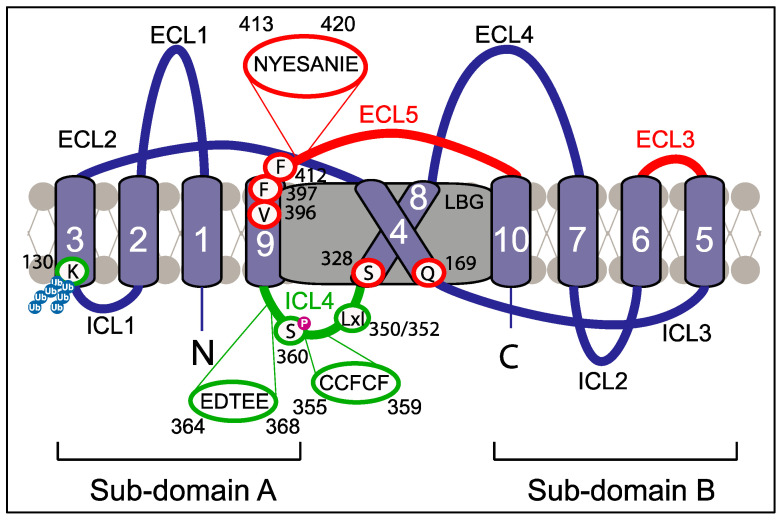
Schematic of domain organization and mapping of functional elements of S5 based on the structure by Pye et al. [30]. Helices and amino acids are numbers in white and black, respectively. Red: determinants for restriction of HIV-1 virion infectivity. Green: determinants for antagonism by HIV-1 Nef. LBG: lipid-binding groove. See text for details.

**Figure 2 viruses-16-00284-f002:**
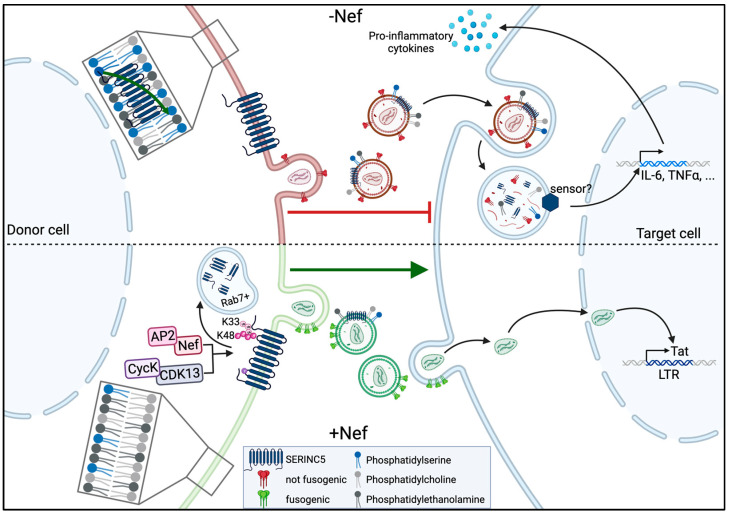
Schematic summarizing the current understanding of restriction to HIV-1 infectivity and sensitization for innate recognition by target cells by SERINC5 and their antagonism from HIV-1 Nef. Depicted is the production of HIV-1 particles from S5-positive cells (donor) and the process of infection of new target cells with cell-free virus particles. In the upper half, the S5 restriction is in place because of a lack of the antagonist Nef; the lower half depicts how Nef counteracts the S5 restriction (see text for details). The figure was generated using Biorender.

**Figure 3 viruses-16-00284-f003:**
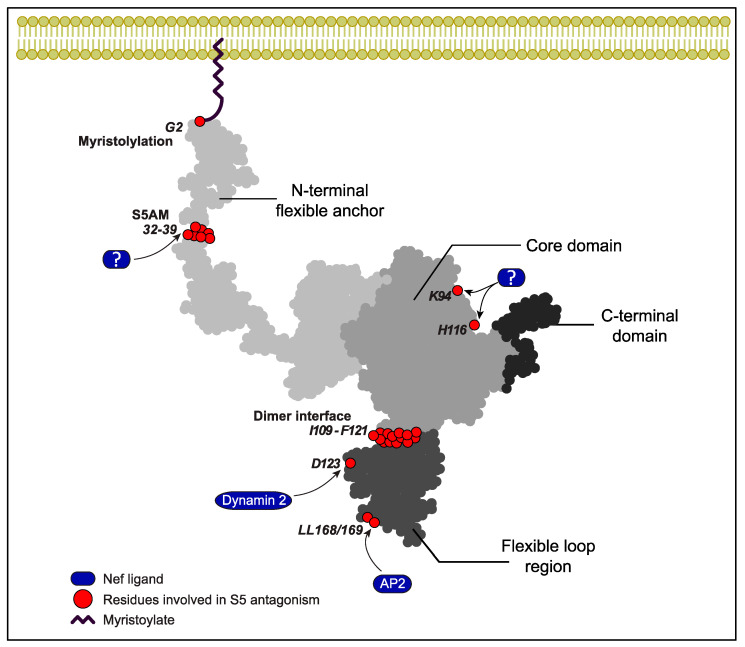
Structural model mapping the determinants of SERINC5 antagonism in HIV-1 Nef. Schematic depiction of the structure of HIV-1 Nef based on the structures of the N-terminus (PDB: 1QA5) and the core and C-terminal flexible loop (2NEF). Amino acid numbering according to their position in HIV-1 Nef NL4-3. Indicated are residues and host cell ligands involved in S5 antagonism (see text for details).

## Data Availability

This review article does not contain new data to make available.

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
