# Peer review of "Beyond Impairment of Virion Infectivity: New Activities of the Anti-HIV Host Cell Factor SERINC5"

_viruses, 2024, doi:10.3390/v16020284_

Round 1
Reviewer 1 Report
Comments and Suggestions for Authors
This review provided an up-to-date account of an important HIV-1 host restriction factor SERINC proteins, mainly S3 and S5. In addition to the well studied viral restriction mechanisms by S3/5 and the antagonism by HIV-1 Nef, recent reports on the phopholipid scramblase activity of SERINC and its role in stimulating antiviral signaling in certain cell types such as myeloid cells were also discussed in the context of SERINC antiviral properties. The figures are informative, facilitate the understanding of the text.
A few minor points for the authors to address.
Abstract, lines 13-14, "Until recently, the mechanisms that mediate restriction as well as antagonism remained largely elusive." This is not an accurate statement, given the related knowledge that is discussed in great depth in sections 4 and 5, i.e much is known.
Line 40, "Intense research in the past decade identified...", considering the report of anti-HIV-1 activity of APOBEC3G in 2002, it has been two decades, so it should be "past decades"
The authors need to explicitly define "restriction factors", are all antiviral factors considered as restriction factors?
Lines 47 - 48, Vif is essential for HIV-1 to grow in CD4+ T cell lines that express APOBEC3G, as well as in PBMCs.
Line 50, "no such antagonistic activity was identified for the Nef protein." Nef antagonization of S3/5 was reported in 2015, almost 8 years ago, this sentence needs to be revised, such as "no such antagonistic activity was identified for the Nef protein until 2015."
Discussion of how other viruses counter S3/5 should help to understand more Nef antagonization of S3/5. As it stands now, the related description is very brief (line 70).
Author Response
This review provided an up-to-date account of an important HIV-1 host restriction factor SERINC proteins, mainly S3 and S5. In addition to the well studied viral restriction mechanisms by S3/5 and the antagonism by HIV-1 Nef, recent reports on the phopholipid scramblase activity of SERINC and its role in stimulating antiviral signaling in certain cell types such as myeloid cells were also discussed in the context of SERINC antiviral properties. The figures are informative, facilitate the understanding of the text.
Reply: We thank the reviewer for the positive overall assessment of our manuscript and the constructive comments below.
A few minor points for the authors to address.
Abstract, lines 13-14, "Until recently, the mechanisms that mediate restriction as well as antagonism remained largely elusive." This is not an accurate statement, given the related knowledge that is discussed in great depth in sections 4 and 5, i.e much is known.
Reply: The sentence was rewritten to take the reviewers comment into account.
Line 40, "Intense research in the past decade identified...", considering the report of anti-HIV-1 activity of APOBEC3G in 2002, it has been two decades, so it should be "past decades"
Reply: Corrected as suggested.
The authors need to explicitly define "restriction factors", are all antiviral factors considered as restriction factors?
Reply: A more precise definition of restriction factors has been added to the text (lines 40-42)
Lines 47 - 48, Vif is essential for HIV-1 to grow in CD4+ T cell lines that express APOBEC3G, as well as in PBMCs.
Reply: This sentence was modified to take the comment of the reviewer into account.
Line 50, "no such antagonistic activity was identified for the Nef protein." Nef antagonization of S3/5 was reported in 2015, almost 8 years ago, this sentence needs to be revised, such as "no such antagonistic activity was identified for the Nef protein until 2015."
Reply: Corrected as suggested
Discussion of how other viruses counter S3/5 should help to understand more Nef antagonization of S3/5. As it stands now, the related description is very brief (line 70).
Reply: This is briefly discussed in paragraph 5 including references to reviews that focus on this aspect.
Reviewer 2 Report
Comments and Suggestions for Authors
In this review Ahmed et al provide a concise and thorough summary of the field of SERINC5 biology as it stands at the moment. They describe in detail the effect of SERINC5 in HIV-1 infection, as well as the interplay between SERINC5 and Nef. They further delve into the latest developments in the field, such as the role of SERINC5 as a scramblase, the impact of SERINC5 in innate immune responses and finally the effect of SERINC5 on provirus transcription. This is a clear, thorough and easy to read summary of the field. There are just a few suggested changes to enhance clarity.
1Page 1, at the end of line 46, the authors should also state that viruses can also modulate the transcription of RFs, as another strategy of counteracting them.
2For lines 47-49 in page 1, please include the needed reference here.
3Page 2, lines 56-60 is a large run-on sentence that needs to be rewritten for clarity.
4Line 101 in page 2, “LBG” not “LGB”.
5Page 4, line 193, “..can indeed can be reduced..”, please correct the phrase for clarity.
6Page 4, line 203, “….S5 virion incorporation have been…”, should be “S5 incorporated in virions have been….”
PPage 5, line 225-226, “…, expression of dynamin 2,…”. This phrase is confusing and unclear, please rewrite.
8Page 6, line 249-251. References on the papers alluded to need to be included. If it is only 2 amino acid motifs as described following that sentence in the manuscript the authors need to rewrite the followsing phrase in line 249 “…several amino acid motifs”, because 2 is not “several”.
9Page 7, line 269. Please write “we determined” and not “we had determined”.
1Page 7, line 271. Please write “Quantitative lipidomics…” and not “However, quantitative lipidomics..”
1In lines 348-351, please include a reference for the Ramdas and Chande manuscript at the end of the sentence.
Comments on the Quality of English LanguageA couple of very minor errors mentioned in the review. Nothing major.
Author Response
In this review Ahmed et al provide a concise and thorough summary of the field of SERINC5 biology as it stands at the moment. They describe in detail the effect of SERINC5 in HIV-1 infection, as well as the interplay between SERINC5 and Nef. They further delve into the latest developments in the field, such as the role of SERINC5 as a scramblase, the impact of SERINC5 in innate immune responses and finally the effect of SERINC5 on provirus transcription. This is a clear, thorough and easy to read summary of the field. There are just a few suggested changes to enhance clarity.
Reply: We thank the reviewer for the positive assessment of our manuscript and for the constructive comments made below.
1 Page 1, at the end of line 46, the authors should also state that viruses can also modulate the transcription of RFs, as another strategy of counteracting them.
Reply: Corrected as suggested.
2 For lines 47-49 in page 1, please include the needed reference here.
Reply: The requested reference was added.
3 Page 2, lines 56-60 is a large run-on sentence that needs to be rewritten for clarity.
Reply: The sentence was rewritten as suggested by the reviewer.
4 Line 101 in page 2, “LBG” not “LGB”.
Reply: Corrected as suggested.
5 Page 4, line 193, “..can indeed can be reduced..”, please correct the phrase for clarity.
Reply: The typo was corrected for clarity.
6 Page 4, line 203, “….S5 virion incorporation have been…”, should be “S5 incorporated in virions have been….”
Reply: Corrected as suggested.
7 Page 5, line 225-226, “…, expression of dynamin 2,…”. This phrase is confusing and unclear, please rewrite.
Reply:The sentence was rewritten for clarity.
8 Page 6, line 249-251. References on the papers alluded to need to be included. If it is only 2 amino acid motifs as described following that sentence in the manuscript the authors need to rewrite the followsing phrase in line 249 “…several amino acid motifs”, because 2 is not “several”.
Reply: The paragraph was edited to improve clarity and the required references were added.
9 Page 7, line 269. Please write “we determined” and not “we had determined”.
Reply: Corrected as suggested.
10 Page 7, line 271. Please write “Quantitative lipidomics…” and not “However, quantitative lipidomics..”1
Reply: Corrected as suggested.
11 In lines 348-351, please include a reference for the Ramdas and Chande manuscript at the end of the sentence.
Reply: The reference was added as suggested.
Reviewer 3 Report
Comments and Suggestions for Authors
This is a thorough review by Ahmed and colleagues on the roles of SERINC5 in host defense. Here, recent findings that expand the functions of this restriction factor are discussed to give a better overview of the relevance of SERINC5 in antiviral defense. The manuscript is well written, all the relevant reports are cited, and the figures efficiently transmit the information the authors intend to convey. I do have some minor suggestions:
1. Abstract. Line 20. I would remove “myeloid” since a recent study reported that SERINC5 can also impact HIV gene expression in non-myeloid cells (PMC10537789). Similarly, I would remove “monocyte-derived” from line 306.
2. Line 67. The authors should cite a publication in which SERINC5 was reported to induce MDA5-dependent type I IFN signaling, which reduced replication of CSFV (PMC7498654)
3. Line 72. Add a “comma” after “host”.
4. Lines 150-151. Sentence “which may explain the lack of S5 depletion on HIV-1 spread”. This reviewer had a hard time understanding what the authors were trying to say here. Please, rephrase.
5. Line 193. “Can indeed can be” please, correct sentence construction.
6. Line 313. I would add the following sentence (or similar) right after MDA5. “This is in line with recent reports showing that SERINC5 interacts with MDA5 to promote type-I IFN signaling (PMC7498654)”.
7. Line 369. The authors should mention that the Shi et al study found that the effect of SERINC5 on HIV gene expression was associated with a defect in proviral DNA integration and that the authors also observed an inhibition in the expression of ectopic DNA, which suggests additional implications for SERINC5 in host defense.
8. Line 366. I would remove “myeloid” from the sentence or indicate that some but not all of the above-described functions of SERINC5 appear to be specific for myeloid cells.
9. Line 377. Add a statement about SERINC5 impacting HIV gene expression as part of these novel mechanisms.
10. Figure 2. Top half. Include a definition for the vesicle depicted (endosome, lysosome?) also, what is the blue hexagon representing?
11.Figure 4. The legend indicates reference 89. However, either the panel (mechanism) or the reference are wrong.
Author Response
- Abstract. Line 20. I would remove “myeloid” since a recent study reported that SERINC5 can also impact HIV gene expression in non-myeloid cells (PMC10537789). Similarly, I would remove “monocyte-derived” from line 306.
Reply: We now specify later in the text that the findings reported by Shi et al were made in HEK293T and Jurkat cells and that these cell lines are not of myeloid origin.
- Line 67. The authors should cite a publication in which SERINC5 was reported to induce MDA5-dependent type I IFN signaling, which reduced replication of CSFV (PMC7498654)
Reply: Thanks for pointing out this oversight, we included this reference.
- Line 72. Add a “comma” after “host”.
Reply: Was added.
- Lines 150-151. Sentence “which may explain the lack of S5 depletion on HIV-1 spread”. This reviewer had a hard time understanding what the authors were trying to say here. Please, rephrase.
Reply: Was rephrased
- Line 193. “Can indeed can be” please, correct sentence construction.
Reply: Was corrected
- Line 313. I would add the following sentence (or similar) right after MDA5. “This is in line with recent reports showing that SERINC5 interacts with MDA5 to promote type-I IFN signaling (PMC7498654)”.
Reply: We added such a statement.
- Line 369. The authors should mention that the Shi et al study found that the effect of SERINC5 on HIV gene expression was associated with a defect in proviral DNA integration and that the authors also observed an inhibition in the expression of ectopic DNA, which suggests additional implications for SERINC5 in host defense.
Reply: We added such a statement.
- Line 366. I would remove “myeloid” from the sentence or indicate that some but not all of the above-described functions of SERINC5 appear to be specific for myeloid cells.
Reply: We now specify that the findings reported by Shi et al were made in HEK293T and Jurkat cells and that these cell lines are not of myeloid origin. 9. Line 377. Add a statement about SERINC5 impacting HIV gene expression as part of these novel mechanisms.
- Figure 2. Top half. Include a definition for the vesicle depicted (endosome, lysosome?) also, what is the blue hexagon representing?
Reply: As annotated in the figures, these represent proinflammatory cytokines and a yet-to-be identified sensor, respectively.
11.Figure 4. The legend indicates reference 89. However, either the panel (mechanism) or the reference are wrong.
Reply: Thanks for spotting this mistake, this was the wrong reference indeed and we corrected it.